# Translation-Invariant Zero-Phase Wavelet Methods for Feature Extraction in Terahertz Time-Domain Spectroscopy

**DOI:** 10.3390/s22062305

**Published:** 2022-03-16

**Authors:** Mahmoud E. Khani, Mohammad Hassan Arbab

**Affiliations:** Biomedical Engineering Department, Stony Brook University, Stony Brook, NY 11790, USA; mahmoud.ebrahimkhani@stonybrook.edu

**Keywords:** maximal overlap discrete wavelet transform, terahertz time-domain spectroscopy, feature extraction, material characterization

## Abstract

Wavelet transform is an important tool in the computational signal processing of terahertz time-domain spectroscopy (THz-TDS) measurements. Despite its prevalence, the effects of using different forms of wavelet transforms in THz-TDS studies have not been investigated. In this paper, we explore the implications of using the maximal overlap discrete wavelet transform (MODWT) versus the well-known discrete wavelet transform (DWT). We demonstrate that the spectroscopic features extracted using DWT can vary over different overlapping frequency ranges. On the contrary, MODWT is translation-invariant and results in identical features, regardless of the spectral range used for its implementation.We also demonstrate that the details coefficients obtained by the multiresolution analysis (MRA) using MODWT are associated with zero-phase filters. In contrast, DWT details coefficients suffer from misalignments originated from the down- and upsampling operations in DWT pyramid algorithm. Such misalignments have adverse effects when it is critical to retain the exact location of the absorption lines. We study the differences of DWT and MODWT both analytically and experimentally, using reflection THz-TDS measurements of α-lactose monohydrate. This manuscript can guide the researchers to select the right wavelet analysis tool for their specific application of the THz spectroscopy.

## 1. Introduction

Terahertz time-domain spectroscopy (THz-TDS) has emerged as a promising technique for non-destructive testing applications such as monitoring hidden defects [1,2,3], non-contact thickness measurement [4,5], investigation of artworks and historical artifacts [6,7], and quantification of polymorphism, crystallinity, and bulk porosity in pharmaceuticals [8,9]. Many substances, including molecular crystals and polar gases, possess distinctive spectroscopic fingerprints in the THz range, which allow for non-invasive material identification, sample characterization, and quality control [10,11,12]. However, electromagnetic scattering can significantly alter and obscure these vibrational absorption features [13,14,15,16]. This phenomena in turn can hinder spectroscopic studies and chemical sensing in the presence of rough surface or volume scattering. Additionally, multiple Mie scattering can introduce resonance-shaped spectral artifacts into the measured THz spectra [17]. To address these challenges, signal processing and experimental techniques have been introduced to retrieve spectroscopic information in the presence of such scattering effects [14,17,18,19,20,21].

The widespread applications of THz-TDS demand utilizing advanced signal processing techniques to enhance the precision and accuracy in identification of spectroscopic features [22,23]. Because THz pulses are inherently sparse [24], and they contain localized features in both time and frequency domains, wavelet transform is particularly advantageous for this purpose. Wavelet transform has been used extensively for denoising time- and Fourier-domain THz signals [25,26,27,28,29,30] in addition to THz images obtained using CCD cameras [31,32]. Moreover, wavelet shrinkage accompanied by a Fourier-domain Wiener deconvolution has been proven to be effective in finding the thickness of optically-thin layers in biological samples [33] and art paintings [34]. THz compressive sensing [35], THz tomographic imaging [36], and THz feature extraction for differentiation between various pathological tissues [37,38] are also among the main applications of the wavelet transform in analyzing THz-TDS measurements.

Recently, in order to enable in-line process control and real-time THz sensing applications, we have designed and fabricated a fiber-coupled THz spectral imager, named Portable HAndheld Spectral Reflection (PHASR) Scanner [39]. This device incorporates the TERA-ASOPS (Asynchronous OPtical Sampling) dual-fiber-laser spectrometer (Menlo Systems, Inc., Newton, NJ, USA) into a collocated, telecentric imaging configuration, which utilizes an f-θ lens [40] and a two-axis motorized scanning system [41]. We have conducted multiple experiments on the utility of the PHASR Scanner for material identification [42] and biomedical imaging [43] in the presence of electromagnetic scattering. We have shown that because wavelet transform provides spectral localization at multiple resolution scales, when applied to THz absorption or reflection coefficients, it can separate the characteristic resonant signatures from the scattering-induced spectral artifacts [15,16,17,44]. Furthermore, the wavelet coefficient levels associated with resonant signatures can be identified using total variation in reflection spectroscopy [16], or bimodality coefficient [17] and principal component analysis [45] in the transmission geometry.

In all aforementioned studies, two main forms of the wavelet transform have been utilized. First, the critically-sampled discrete wavelet transform (DWT), and second, an undecimated form of the DWT, named the maximal overlap DWT (MODWT) [46]. In this paper, we study the implications of using MODWT and DWT for feature extraction in THz spectroscopic data. We demonstrate two main advantages of MODWT over DWT, including MODWT invariance with respect to the frequency range and association of MODWT multiresolution analysis (MRA) with zero-phase filters. These advantages come at a slightly higher computational cost, which is affordable for many applications due to the high efficiency in modern computer processing units. In the following, we first explain the differences between DWT and MODWT analytically. Then, we describe the experimental methodology in our study to obtain the THz-TDS measurements. We will show the effects of using MODWT versus DWT in extracting the resonances of α-lactose over the spectral region of f=0.3–1.6 THz. These results elucidate the advantages of MODWT in sensing applications, where it is critical to retain the exact location of the spectral features, such as a resonant frequency.

## 2. Materials and Methods

### 2.1. Analytical Description

#### 2.1.1. Translation Invariance

The DWT decomposes a signal into a set of wavelet and scaling coefficient vectors. Each vector describes the localized changes in the signal at a specific variation scale. The scale is an interval of the signal over which weighted averages of that signal (i.e., the scaling coefficients) and the differences of those weighted averages (i.e., the wavelet coefficients) are calculated. The DWT coefficients are obtained using the pyramid algorithm shown in Figure 1a [46].

Here, x(n) is the input signal, which can be a THz pulse either in the time or Fourier domain. The $w_{j}\left(n\right)$ is the vector of the *j*th-level wavelet coefficients of x(n). It is associated with differences in the weighted averages of x(n) over the scale of $\tau_{j}=\delta n2^{j-1}$, where δn indicates the sampling period in x(n). Equivalently, $w_{j}\left(n\right)$ corresponds to the spectral content of x(n) in the range of $\left[1/\left(2^{j+1}\delta n\right):1/\left(2^{j}\delta n\right)\right]$, displaying band- or high-pass characteristics [46]. The $v_{j}\left(n\right)$ is composed of the *j*th-level scaling coefficients of x(n). It is associated with the weighted averages of x(n) over the scale of $\tau_{j+1}=\delta n2^{j}$, or, equivalently, it corresponds to the spectral content of x(n) in the range of $\left[0:1/\left(2^{j+1}\delta n\right)\right)$, displaying low-pass characteristics. The *j*th-level of the DWT pyramid algorithm is given by [46],
(1)$$w_{j}\left(n\right)=\sum_{k=0}^{K-1}h\left(k\right)v_{j-1}\left(\left(2n+1-k\right)\;\mathrm{mod}\;\left(N/2^{j-1}\right)\right);\;n=0,\ldots ,N/2^{j}-1,$$
and
(2)$$v_{j}\left(n\right)=\sum_{k=0}^{K-1}g\left(k\right)v_{j-1}\left(\left(2n+1-k\right)\;\mathrm{mod}\;\left(N/2^{j-1}\right)\right);\;n=0,\ldots ,N/2^{j}-1,$$
yielding the wavelet and scaling coefficients, respectively. In Equations (Equation 1) and (Equation 2), *h* and *g* are the mother wavelet and scaling filters of size *K*. It should be noted that because of downsampling by two, the number of DWT wavelet and scaling coefficients at level *j* is $\frac{1}{2^{j}}$-th of the length of x(n), *N*. This progressively coarser sampling in DWT is advantageous for applications that rely on compact signal representations, such as signal and image compression [47,48]. However, it leads DWT coefficients to be critically sensitive to the range of analysis. According to Equations (Equation 1) and (Equation 2), $w_{j}$ and $v_{j}$ are formed by discrete, cyclic convolutions of *h* and *g* with $v_{j-1}$, respectively, and the results are downsampled by two. The convolution of two discrete functions (e.g., *h* and $v_{j-1}$) describes the summation of the product of two functions after one is reversed ($v_{j-1}\left(-k\right)$) and shifted ($v_{j-1}\left(n-k\right)$), over all possible values of shifts (*n*). In Equations (Equation 1) and (Equation 2), the product of *h* and $v_{j-1}$ or *g* and $v_{j-1}$ depends on which data points in $v_{j-1}$ are excluded in downsampling.

Conversely, in the MODWT pyramid algorithm, which is shown by the block diagram in Figure 1b, the downsampling operation is excluded. In other words, the wavelet and scaling coefficients at all decomposition levels have the same size as the original signal. We use the symbol “∼” to differentiate MODWT wavelet and scaling filters and coefficients (${\tilde{h}}_{j}$, ${\tilde{g}}_{j}$, ${\tilde{w}}_{j}$, and ${\tilde{v}}_{j}$) from DWT. The MODWT wavelet and scaling filters, $\tilde{h}$ and $\tilde{g}$, are related to *h* and *g* by $\tilde{h}=h/\sqrt{2}$ and $\tilde{g}=g/\sqrt{2}$, respectively [46]. In the block diagram of Figure 1b, the MODWT wavelet and scaling filters at level *j* are upsampled by $2^{j-1}$ with respect to $\tilde{h}$ and $\tilde{g}$ to form a multiresolution decomposition. The upsampling by $2^{j-1}$ is achieved by inserting $2^{j-1}-1$ zeros between successive elements of $\tilde{h}$. For example, if $\tilde{h}=\left[\tilde{h}\left(0\right),\tilde{h}\left(1\right),\ldots ,\tilde{h}\left(K-1\right)\right]$ is a MODWT wavelet filter with *K* elements, the second-level wavelet filter is formed by ${\tilde{h}}_{2}=\left[\tilde{h}\left(0\right),0,\tilde{h}\left(1\right),0,\ldots ,0,\tilde{h}\left(K-1\right)\right]$, composed of 2K−1 elements. Accordingly, the *j*th-level of the MODWT pyramid algorithm is given by [46],
(3)$${\tilde{w}}_{j}\left(n\right)=\sum_{k=0}^{K-1}\tilde{h}\left(k\right){\tilde{v}}_{j-1}\left(\left(n-2^{j-1}k\right)\right)\;\mathrm{mod}\;N);\;n=0,\ldots ,N-1,$$
and
(4)$${\tilde{v}}_{j}\left(n\right)=\sum_{k=0}^{K-1}\tilde{g}\left(k\right){\tilde{v}}_{j-1}\left(\left(n-2^{j-1}k\right)\right)\;\mathrm{mod}\;N);\;n=0,\ldots ,N-1.$$
Importantly, the convolution sums in Equations (Equation 3) and (Equation 4) include all possible products of the upsampled $\tilde{h}$ and ${\tilde{v}}_{j-1}$ or upsampled $\tilde{g}$ and ${\tilde{v}}_{j-1}$. Therefore, the MODWT wavelet and scaling coefficients are translation-invariant and do not depend on the range of the analysis.

#### 2.1.2. Zero-Phase Filtering

The phase functions of the DWT and MODWT filters change the position of the extracted features in the wavelet domain with respect to their locations in the original signal. If the phase function of a filter is linear, shifting its output coefficients by a certain number, which is equal to the slope of its phase as a function of frequency, will compensate the introduced misalignments [15]. However, the phase functions of the orthogonal wavelet and scaling filters, which are used in DWT and MODWT, are not linear [49]. Moreover, the recursive filtering of a signal by the filters in the pyramid algorithm makes the non-linearity of the phase functions more complicated. To address this problem in applications where retaining the location of the features is important, the cross-correlation of wavelet and scaling coefficients with their corresponding wavelet and scaling filters can be used to form a zero-phase multiresolution analysis. In the DWT pyramid algorithm of Figure 1a, the *j*th-level wavelet coefficients are formed by j−1 convolutions between x(n) and g(n) and a final convolution with h(n). Equivalently, we can build *j*th-level DWT wavelet and scaling filters given by $h_{j}=\overset{j-1}{\overset{︷}{g⊛g⊛\ldots g}}⊛h$ and $g_{j}=\overset{j}{\overset{︷}{g⊛g⊛\ldots g}}$, respectively, where the symbol $“{⊛}^{”}$ indicates a cyclic convolution. Thereafter, $w_{j}$ and $v_{j}$, which were previously calculated using Equations (Equation 1) and (Equation 2), can be directly obtained from x(n) given by [46],
(5)$$w_{j}\left(n\right)=\sum_{k=0}^{K_{j}-1}h_{j}\left(k\right)x\left(\left(2^{j}\left(n+1\right)-1-k\right)\;\mathrm{mod}\;\left(N\right)\right);\;n=0,\ldots ,N/2^{j}-1,$$
and
(6)$$v_{j}\left(n\right)=\sum_{k=0}^{K_{j}-1}g_{j}\left(k\right)x\left(\left(2^{j}\left(n+1\right)-1-k\right)\;\mathrm{mod}\;\left(N\right)\right);\;n=0,\ldots ,N/2^{j}-1,$$
where $K_{j}=\left(2^{j}-1\right)\left(K-1\right)+1$ is the length of $h_{j}$ and $g_{j}$. Consequently, the aforementioned cross-correlation between $h_{j}$ and $w_{j}$ is given by [46],
(7)$$d_{j}\left(n\right)=h_{j}⍟w_{j}^{↑2^{j}}\left(n\right)=\sum_{k=0}^{K_{j}-1}h_{j}\left(k\right)w_{j}^{↑2^{j}}\left(\left(n+k\right)\;\mathrm{mod}\;N\right),$$
where $w_{j}^{↑2^{j}}$ is the *j*th-level wavelet coefficients upsampled by $2^{j}$, which is formed by inserting $2^{j}-1$ zeros before every element of $w_{j}$. The symbol “⍟” in Equations (6) and (7) indicates a cyclic cross-correlation. The *d_j_* is composed of the *j*th-level DWT details coefficients and, similar to *w_j_*, it is associated with differences in the weighted averages of *x* over the scale of *τ_j_ = δn*2^*j*−1^. Assuming that *X*(*f*), *D_j_*(*f*), and *H_j_*(*f*) are used to represent the Fourier transform of *x*(*n*), *d_j_*(*n*), and *h_j_*(*n*), respectively, according to the convolution theorem and while ignoring the down- and upsampling operations, *D_j_*(*f*) is obtained by
(8)$$\begin{array}{c} D_{j}\left(f\right)=X\left(f\right)H_{j}\left(f\right)H_{j}^{*}\left(f\right)=X\left(f\right){\left|H_{j}\left(f\right)\right|}^{2}, \end{array}$$
where |*H_j_*(*f*)|^2^ has a zero phase function. Therefore, it only modifies the magnitude spectrum of *X*(*f*), while its phase spectrum remains intact. Thus, there will be no misalignments between *d_j_* and *x* caused by the filters’ phase functions. However, it should be noted that the downsampling of DWT wavelet coefficients in Equation (1) and their subsequent upsampling in Equation (7) can still introduce mismatches between the location of the features in *d_j_* and *x*.

Similar to DWT, MODWT details coefficients are obtained by a cross-correlation between ${\tilde{h}}_{j}$ and ${\tilde{W}}_{j}$ given by [46],
(9)$${\tilde{d}}_{j}\left(n\right)={\tilde{h}}_{j}⍟{\tilde{w}}_{j}\left(n\right)=\sum_{k=0}^{K_{j}-1}{\tilde{h}}_{j}\left(k\right){\tilde{w}}_{j}\left(\left(n+k\right)\;\mathrm{mod}\;N\right),$$
or, equivalently in the Fourier domain,
(10)$${\tilde{D}}_{j}\left(f\right)=X\left(f\right){\tilde{H}}_{j}\left(f\right){\tilde{H}}_{j}^{*}\left(f\right)=X\left(f\right){\left|{\tilde{H}}_{j}\left(f\right)\right|}^{2}.$$
Therefore, ${\tilde{d}}_{j}$ is also formed by zero-phase filtering of X(f) with $|{\tilde{H}}_{j}{\left(f\right)|}^{2}$. Moreover, because the MODWT wavelet coefficients are neither down- nor upsampled, there will be no misalignments between ${\tilde{d}}_{j}$ and *x*.

### 2.2. Measurement Setup

The THz–TDS measurements are obtained using a TERA-ASOPS (ASynchronous OPtical Sampling) high-speed THz time-domain spectrometer (Menlo Systems, Inc., Newton, NJ, USA). The ASOPS system is composed of two 1560 nm mode-locked femtosecond lasers, which are slightly detuned in their repetition rates by a fixed value of Δf=100 Hz. The difference in the repetition rate results in pulses from the two lasers that are emitted at $t_{0}$ to be separated at detector by Δt given by,
(11)$$\Delta t=|\frac{1}{f_{\mathrm{probe}}}-\frac{1}{f_{\mathrm{pump}}}|=\frac{\Delta f}{f_{\mathrm{probe}}.f_{\mathrm{pump}}},$$
where $f_{\mathrm{pump}}=100$ MHz in our experiments. This Δt enables the probe pulse to linearly sample the THz pulse generated by the pump pulse. Figure 2a shows a schematic of the measurement setup. The femtosecond pump and probe pulses are guided by optical fibers to the emitter and detector photoconductive antennae (PCA), generating THz radiation with an effective bandwidth of f=0.3–1.6 THz. Two TPX50 lenses (Menlo Systems, Inc., Newton, NJ, USA) collimate and focus the generated beams on the sample surface at an angle of incidence of $\theta_{i}=35^{\circ }$ and a focal spot size of approximately 1.2 mm at the peak frequency of the THz emissions. Similar lenses collimate and refocus the specular reflections on the detector antenna.

### 2.3. Sample Preparation

We prepared sample disks made from α-lactose monohydrate powders (Spectrum Chemical Mfg. Corp., Gardena, CA, USA) with resonant frequencies at 0.53, 1.2, and 1.4 THz [50,51]. To create the sample disks, we mixed the α-lactose powders with ultra-fine high-density polyethylene (HDPE) (1:1 ratio), and pressed the mixture under 3000 PSI load for about 30 min, yielding pellets with approximately 4 mm thickness and 50 mm diameter. Therefore, each sample was thick enough to avoid any overlaps between the front- and back-reflections.

### 2.4. Signal Conditioning

The main pulse in the measured time-domain electric field is windowed to exclude a second Fabry Perot reflection pulse. To avoid creation of an artificial step in the signal, we apply a tapering Hann window to the data points before the cut-off location. To account for the spatial heterogeneity in the samples, the average amplitude spectrum of ten disjoint measurements over the sample surface is used. Additionally, the sample measurements are deconvolved in the Fourier domain by the reflections from a reference mirror placed at the location of the sample surface. The amplitude spectrum of a sample with a smooth surface exhibits a linear trend with a close-to-zero slope. The effect of this trend in each amplitude spectrum is removed using linear curve fitting. Following this step, tapering Hann windows are applied at the boundary points in the frequency-domain to smooth the transition into the baseline. The spectral window range is chosen such that it does not affect the signal within the useful bandwidth. For a DWT with *J* levels of decomposition, the size of the signal *N* should be an integer-multiple of $2^{J}$. Therefore, we zero-pad the signals to $\left\lceil \frac{N}{2^{J}}\right\rceil 2^{J}$ for a DWT with J=8 levels of decomposition. Subsequently, the DWT and MODWT wavelet and details coefficients are calculated using Equations (Equation 1), (Equation 3), (Equation 7), and (Equation 9). We use the MATLAB software (Mathworks, Natick, MA, USA) for analyzing the data and creating the figures.

## 3. Results

Figure 2b shows an example specular reflection measured from an α-lactose pellet. Figure 2c shows the spectral amplitude (SA) of α-lactose over the frequency range of f=0.3–1.13 THz. As described in the “signal conditioning” section, to obtain the SA, the sample measurement is deconvolved in the Fourier domain using a reference measurement, and the result is detrended using linear curve-fitting. The zero-crossing of the resonant signature in the SA signal, which is delineated using a red arrow in Figure 2c, is positioned at 0.53 THz. Figure 2d shows the vertically-offset fourth-level DWT wavelet coefficients of the SA signal. The wavelet coefficients are calculated using the least asymmetric mother wavelet with four vanishing moments, i.e., the LA(8) or sym4 wavelet filter [49]. Each trace in Figure 2d is composed of the wavelet coefficients calculated over different overlapping spectral ranges of SA. These ranges include f=0.30–1.10 THz (purple line), f=0.31–1.11 THz (red line), f=0.32–1.12 THz (blue line), and f=0.33–1.13 THz (green line). The beginning frequency of each trace is delineated using a vertical line. It can be seen that the features extracted by DWT wavelet coefficients depend on the spectral range used for calculation of the wavelet coefficient. In particular, with the beginning frequency at f=0.30 and f=0.32 THz in the purple and red traces, the wavelet coefficients do not extract the resonance of α-lactose. On the contrary, the same resonant feature can be captured by the red and green traces with the beginning frequency at f=0.31 and f=0.33 THz, respectively. In Figure 2d, each box with the same color as the corresponding trace shows the overlap between the wavelet filter (dashed line) and the resonant signature (solid line) at 0.53 THz. It can be observed that, because of the downsampling operation in Equation (Equation 1), the wavelet filter and the resonant signature overlap in neither of the purple and blue boxes. However, there is a perfect overlap between the wavelet filter and the resonant signature in both of the red and green boxes, leading to the capturing of the resonance.

Figure 3 compares the SA of α-lactose as the control sample in Figure 3a with the vertically-offset fourth-level MODWT wavelet coefficients in Figure 3b. Similar to Figure 2d, each trace in Figure 3b is composed of the wavelet coefficients calculated over a different frequency range, including f=0.30–1.10 THz in the purple trace, f=0.31–1.11 THz in the red trace, f=0.32–1.12 THz in the blue trace, and f=0.33–1.13 THz in the green trace. It can be observed that, unlike DWT, all MODWT wavelet coefficients, regardless of their frequency range, can extract the resonant signature. In Figure 3b, each box is shaded with the same color as the corresponding wavelet coefficient trace. The inset boxes display the overlap between the MODWT wavelet filter and the resonant signature at 0.53 THz. It can be seen that the wavelet filter and the resonant signature overlap in all the boxes, leading to the extraction of the resonant feature by their corresponding traces.

It should be noted that to align the wavelet coefficients in Figure 2d and Figure 3b with the SA signal, wavelet coefficients are circularly shifted, following the phase function correction approach described in [15]. The number of these circular shifts is given by the power of *T* in Figure 2d and Figure 3b, i.e., $T^{-3}W_{4}$ and $T^{-53}{\tilde{W}}_{4}$, where a negative sign indicates that the circular shifts are to the left. On the contrary, as we explain in the “zero-phase filtering” section, details vectors are inherently associated with zero-phase filters, while they have the same band- or high-pass characteristics as the wavelet coefficients. Therefore, they do not require the phase corrections for the purpose of aligning the features. Figure 3c compares the fourth-level DWT and MODWT wavelet and details coefficients. To better demonstrate the impact of the phase function on alignment of the features, wavelet coefficients in Figure 3c are not circularly shifted. It can be observed that $W_{4}$ and ${\tilde{W}}_{4}$ do not align at the resonant frequency, which is delineated using a vertical dashed line. In contrast, the resonant features extracted by details coefficients are aligned at 0.53 THz without requiring any phase correction. Furthermore, to validate the superiority of MODWT over DWT details, Figure 3d–f compare the third-level DWT and MODWT details coefficients of the α-lactose’s SA. In Figure 3d, the SA signal is shown over the entire measurement bandwidth of f=0.3–1.6 THz. The resonances of α-lactose at 0.53, 1.2, and 1.4 THz are delineated using vertical dashed lines. Although the details coefficients in Figure 3e do not require any circular shifts, zooming in at the details coefficients around 1.4 THz, which are shown in Figure 3e, reveals that DWT details are slightly misaligned with respect to the resonant frequency. This misalignment arises from the down- and upsampling operations used in calculating the DWT details coefficients. Therefore, in spectroscopic applications where retaining the exact location of features is critical, MODWT details are preferable over DWT. In our previous papers, we have also demonstrated that the resonant signatures of different characteristics, such as spectral height and width, can be identified using wavelet coefficients at different decomposition levels [16,17]. We have presented two computational techniques, i.e., the total variation of second-order derivatives of the reflection coefficient [16] or the bimodality coefficient spectrum of the absorption coefficient [17], to identify the decomposition levels associated with each resonant feature.

Figure 4a,b compares the translation-invariance property between the DWT and MODWT details coefficients. Similar to Figure 2d and Figure 3b, each trace in Figure 4a,b is calculated over a different frequency range. It can be seen that DWT details calculated over f=0.30–1.10 THz (purple trace) and f=0.32–1.12 THz (blue trace) do not contain the α-lactose’s resonance. In contrast, all MODWT details coefficients, regardless of their frequency range, can extract the resonant signature. Finally, Figure 4c–f show the amplitude of the DWT and MODWT details vectors calculated over two different, overlapping frequency ranges down to four levels of decomposition. The horizontal white dashed lines separate the levels of decomposition. The details coefficients at each panel are min–max-normalized between zero and one. Figure 4c,d compare the DWT details coefficients calculated over f=0.3–1 THz and f=0.4–1.1 THz, respectively. It can be seen that the DWT coefficients over the two regions are not alike. Moreover, the resonant lines at 0.53 THz are not clearly distinguishable from the other lines surrounding them. Similarly, Figure 4e,f compare the MODWT details coefficients calculated over the same frequency ranges as Figure 4c,d. The MODWT details coefficients calculated over the two different regions are identical, indicating the translation-invariance of MODWT at all levels of decomposition. Moreover, the resonant lines at both frequency regions are clearly distinct from the other coefficients.

## 4. Conclusions

In this paper, we studied the main advantages of MODWT over DWT for the extraction of characteristic resonances using broadband terahertz spectroscopy technique. The first advantage is the invariance property of MODWT with respect to the spectral range of analysis. We showed that by changing the frequency range in the DWT analysis of the reflection spectral amplitudes of α-lactose, the wavelet coefficients calculated over different ranges are not alike. Therefore, the characteristic resonances do not appear in the DWT wavelet and details coefficients of all frequency ranges. In particular, we showed that the α-lactose’s resonance at 0.53 THz is not captured by the DWT wavelet and details coefficients over the overlapping spectral ranges of f=0.30–1.10 THz and f=0.32–1.12 THz. However, in MODWT, there is no dependency on the the frequency range, and all wavelet and details coefficients can extract the resonant signature. The second advantage is the zero-phase filtering inherent in calculating the MODWT details coefficients, which results in a perfect alignment between the spectroscopic features in detail coefficients and the original signal. We demonstrated that although DWT details coefficients are also associated with zero-phase filters, because of the down- and upsampling operations used in their calculation, slight mismatches still appear between the details coefficients and the original signal. Our experimental results show that DWT details coefficients associated with α-lactose’s resonant signature at 1.40 THz do not align with the original signal at the resonant frequency, whereas there is a perfect alignment between MODWT details vector and the original spectrum. Due to these advantages, MODWT is a more reliable analytical tool for spectroscopic applications that rely on the extraction and the exact spectral location of the characteristic absorption lines. 

## Figures and Tables

**Figure 1 sensors-22-02305-f001:**
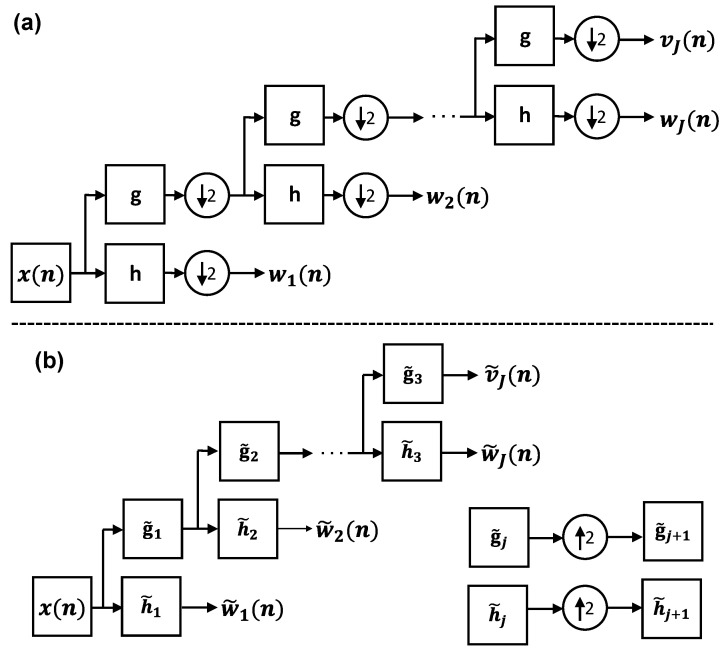
(**a**) The block diagram of the DWT pyramid algorithm. The input signal x(n) is filtered recursively by a set of high-pass wavelet and low-pass scaling filters. The output of each filter is downsampled by a factor of two. (**b**) The block diagram of the MODWT pyramid algorithm. In MODWT, the sampling rate is the same at all levels of decomposition. However, the (j+1)th-level filters are upsampled by a factor of two with respect to the *j*th-level filters.

**Figure 2 sensors-22-02305-f002:**
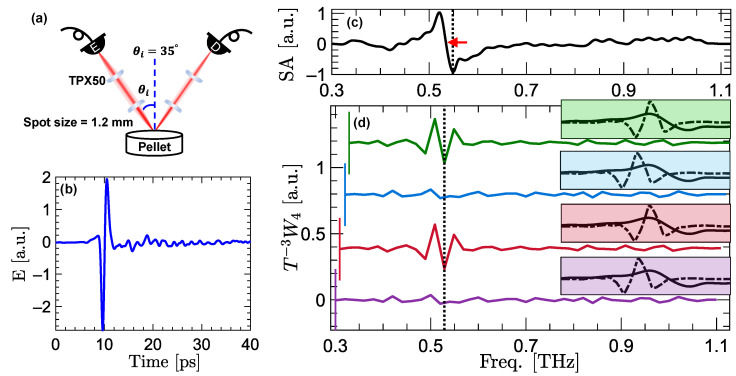
(**a**) The schematic of the measurement setup. The specular reflections are measured at $\theta_{i}=\theta_{r}=35^{\circ }$. (**b**) An example THz pulse reflected from an α-lactose pellet. (**c**) The spectral amplitude (SA) of α-lactose over the frequency range of f=0.3–1.13 THz. A red arrow marks the zero-crossing at 0.53 THz, which is a characteristic resonance mode of α-lactose. (**d**) The vertically-offset fourth-level DWT wavelet coefficients of SA, applied in the frequency domain. Each line in (**d**) is composed of the wavelet coefficients calculated from a different spectral range, i.e., f=0.30–1.10 THz in the purple trace, f=0.31–1.11 THz in the red trace, f=0.32–1.12 THz in the blue trace, and f=0.33–1.13 THz in the green trace. The lower bound of the frequency ranges are emphasized using vertical lines at the beginning of each trace. Each inset box shaded with the same color shows the overlap between the wavelet filter and the resonant signature at 0.53 THz in the corresponding reflection spectra. In each case, it can be noted that whether the resonance at 0.53 THz is resolved in $T^{-3}W_{4}$ depends on the overlap between the spectral feature and the DWT wavelet filter.

**Figure 3 sensors-22-02305-f003:**
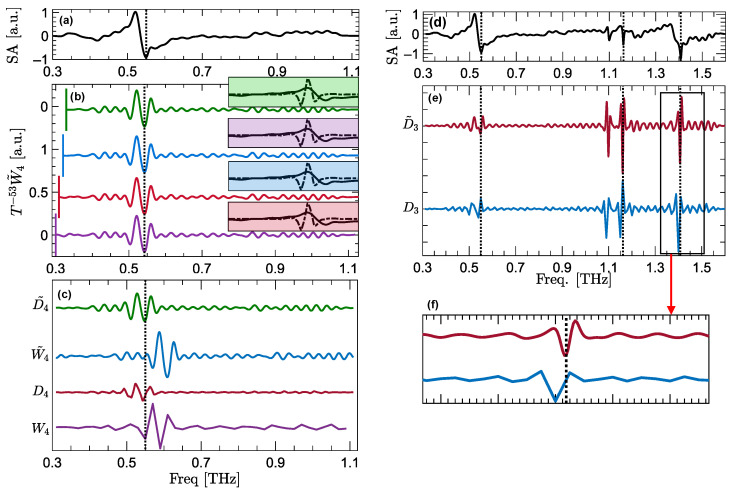
(**a**) The SA of α-lactose over the frequency range of f=0.3–1.13 THz as the control sample. (**b**) The vertically-offset fourth-level MODWT wavelet coefficients of the SA signal, applied in the frequency domain. Each line in (**b**) is calculated over a different frequency range, including f=0.30–1.10 THz in the purple line, f=0.31–1.11 THz in the red line, f=0.32–1.12 THz in the blue line, and f=0.33–1.13 THz in the green line. The lower bound of the frequency ranges are emphasized using vertical lines at the beginning of each trace. Each inset box, shaded with the same color, shows the overlap between the wavelet filter (dashed line) and the resonant signature (solid line) at 0.53 THz in the corresponding reflection spectra. (**c**) The vertically-offset fourth-level DWT and MODWT wavelet and details coefficients of the SA signal. Panel (**d**) shows the SA of α-lactose over the entire measurement bandwidth of f=0.3–1.6 THz. In (**e**) are the vertically-offset third-level DWT and MODWT details vectors of the SA signal in (**d**). Panel (**f**) zooms into the details coefficients around 1.4 THz, revealing that the DWT details vector (blue line) is slightly misaligned with respect to the resonant frequency, while MODWT details vector (red line) maintains the exact spectral location. The vertical dashed lines in (**a**–**f**) delineate the resonances of α-lactose.

**Figure 4 sensors-22-02305-f004:**
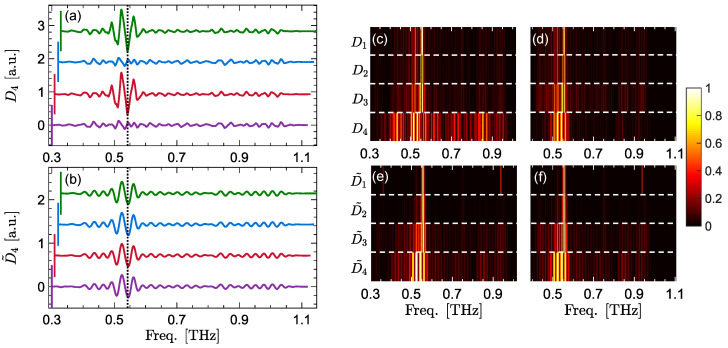
The vertically-offset fourth-level DWT details coefficients, (**a**), are compared to the the same vertically-offset fourth-level MODWT details coefficients in (**b**). Each trace in (**a**,**b**) is calculated over a different frequency range, including f=0.30–1.10 THz in the purple traces, f=0.31–1.11 THz in the red traces, f=0.32–1.12 THz in the blue traces, and f=0.33–1.13 THz in the green traces. In (**c**,**d**) the amplitude of all DWT details coefficients are calculated over f=0.3–1 THz, (**c**), and f=0.4–1.1 THz, (**b**) for a decomposition level of J=4. In (**e**,**f**), the amplitude of all MODWT details coefficients are calculated over f=0.3–1 THz and f=0.4–1.1 THz, respectively, for a decomposition level of J=4. The horizontal white dashed lines separate the levels of decomposition. The details coefficients at each panel are min–max-normalized between zero and one.

## Data Availability

The data presented in this study are available on request from the corresponding author.

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
