# Peer review of "Translation-Invariant Zero-Phase Wavelet Methods for Feature Extraction in Terahertz Time-Domain Spectroscopy"

_sensors, 2022, doi:10.3390/s22062305_

Round 1
Reviewer 1 Report
Authors introduced interesting effects of using different forms of wavelet transforms in terahertz time-domain spectroscopy measurement. The results data is very clear to be understood. In addition, description of the theoretical approaches and analysis is very clear to be understood. There are no English grammar mistakes. Literature search information around 40 references in the introduction section is very good. It is hard to find some errors in the entire manuscript. Therefore, the manuscript can be minor revision with suggestive comments as below.
1. In Line 77, Eqs. -> Equations.
2. Authors had better show the important measured data in the conclusion section.
3. Please provide the information of the company (TERA-ASOPS) such as city and country information.
Reviewer 2 Report
The authors are presenting a comparative study for computational THz-TDS signal processing based on diverse wavelet transform algorithms. The authors show proof the advantages, MODWT technique by extracting the spectral fingerprints from the experimentally measured time domain signals acquired for an a-lactose sample.
The work is well organized and comprehensively described. The introduction provides sufficient background and includes relevant references. The concept is clearly presented and well explained with the supplementary materials. The given work would be very useful for the scientific community.
Even if the paper is already complete in its present form, some concepts must be better explained. Some abbreviations used are misleading and must be corrected throughout the text. I would recommend publication after a minor correction.
I would like to thank all the Authors for their efforts and I kindly ask them to address my comments and suggestions below:
- The text is clearly written and well presented. The English language and style are fine still an overall spell check would be beneficial.
- In figure 2.a. the given scheme is not a specular reflection geometry. It is simply a standard theta-theta reflection geometry at 35 degree of angle of incidence. I ask the authors to correct the specular reflection abbreviation used throughout the text.
- I kindly ask the authors to better identify, if the main peak of the terahertz signal is cut prior to the FFT analysis. Further comment is needed, if a post processing is applied to the acquired electric field
spectrums, in terms of the phase interpolation(prior to unwrapping), and if any smoothing or cutting/zero-pad procedure applied prior to the frequency domain analysis. - The claimed resonance finger prints for the lactose sample must be strengthen with relevant references from literature.
- In line 136: “The ripples following the main THz pulse are caused by the inter-molecular absorption modes in a-lactose and will appear as resonant signatures in the Fourier spectrum.” . the ripples not only include the effect from the possible resonances but also includes the high frequency components of the main E-Field. I kindly ask the authors to further comment on how the filtration effect the main signal(reference signal), frequency response.
- As the proposed technique extracts parameters from the spectral tail of the THz time domain signal, modifying the frequency response diversely at different frequency intervals The possible resonances(spectral finger prints) lay in a large frequency band. I ask the authors to better explain and further comment on the possible side effect on the resonance structure identification.
Wtih my best regards,
Reviewer 3 Report
The paper “Translation-invariant Zero-phase Wavelet Methods for Feature Extraction in Terahertz Time-domain Spectroscopy”, by Mahmoud E. Khani and M. Hassan Arbab, presents results concerning wavelet function utility in the frame of terahertz time domain spectroscopy. The paper is well written, and the information is clear presented.
The article can be accepted for Publication after Minor Changes. My comments are below.
- Abstract- suggestion for the authors: it seems that your abstract is full with name shortcuts, e.g., DWT, etc. The shortcuts were presented with their long name; thus, this thing should be inserted inside the Introduction, not in the abstract.
- Figure 1: the representation is original? If not, please insert the reference.
- I saw the used equations, and the resulted graphs, thus, in which software package were compiled?
- For all presented equations, please insert the literature references.
